# Self-Paced Learning Enhanced Physics-informed Neural Networks for Solving Partial Differential Equations

## Abstract

There is a hit discussion on solving partial differential equation by neural network. The famous PINN (physics-informed neural networks) has drawn worldwide attention since it was put forward. Despite its success in solving nonlinear partial differential equation, the difficulty in converging and the inefficiency in training process are definitely huge concerns. Normally, data for PINN is randomly chosen for a given distribution. Additionally, it's fitted to a model in a meaningless way. Curriculum Learning is a learning strategy that trains a model from easy samples to hard ones, which represents the meaningful human learning order. Self-paced Learning (SPL) is one of the significant branches of Automatic Curriculum Learning, which takes example-wise the training loss as Difficulty Measurer. SPL is an efficient strategy in enhancing the convergence rate of numerous models. In this paper, we propose a novel SPL-PINN learning framework, with SPL to accelerate the convergence progress of PINN. We demonstrate the effectiveness of SPL-PINN in a typical parabolic equation and Burgers equation.

## 1 Introduction

There is an increasing number of methods for solving partial differential equation. Weinan & Yu (2018) proposed a deep learning-based method, the Deep Rize Method, for solving variational problems, which are derived form partial differential equations. Sirignano & Spiliopoulos (2018) proposed a mesh-free method, the Deep Galerkin Method, to approximate the solution of a partial differential equation by a neural network instead of a linear combination of bias functions. (RAI, 2019) aimed to approximate the solution by neural network as well as the inner physical information. There is much profound research related to PINN, the research topics can be filed to three fields: the application of PINN to a specific problem, the parallel PINN and the acceleration of PINN. Papers related to PINN abound, suggesting that a great burst of creativity has occured since the emergence of PINN. In this paper, we concentrate on how to accelerate the converging rate in the training process. Xiang et al. (2021) proposed a self-adaptive loss function which automatically assigns the weights of different objectives. LIU (2021) is also proposed to automatically adjust the weights of different objectives. Thanasutives et al. (2021) treated it as a multi-task learning problem, they proposed adversarial learning to generate high-loss samples, similarly distributed to the original training distribution. Kim et al. (2020) proposed a dynamic pulling method, enabling PINN to learn dynamics of the governing equations. Yu et al. (2022) assumed that the solution of the given partial differential equation is smooth, so they took the gradient of the PDE residual as part of the loss function.Chiu et al. (2021) proposed a CAN-PINN framework, taking the advantage of automatic differentiation and numerical differentiation, increase the accuracy of PINN. Bischof & Kraus (2021) treated it as a multi-objective problem, proposed a novel self-adaptive loss balancing of PINN with random lookback. Generally, the methods aiming at the inefficiency in training process are treating the weights of different losses. In this paper, we concentrate on the training pattern, fit model form easy samples to hard samples.

## 2 RELATED WORK

### 2.1 PHYSICS-INFORMED NEURAL NETWORK

Physics-informed neural network(RAI, 2019) is a successful mesh-free method for solving partial differential equations. Let's start by focusing on a general partial differential equation with both initial-value condition and boundary-value condition.

$$
\begin{aligned}
&\partial_t u(t, x) + \mathcal{L}_x u(t, x) = f(t, x) \quad (x, t) \in \Omega \times (0, T) \\
&u(0, x) = g(x) \quad x \in \Omega \\
&u(t, x) = h(t, x) \quad (x, t) \in \partial\Omega \times (0, T)
\end{aligned}
\tag{1}
$$

Here $u(x, t)$ is the solution of this initial/boundary problem. $L$ is the differential operator on the variable $x$. $\Omega$ is a subset of $R^n$. PINN approximates $u(x, t)$ by neural network with the physical information instead of traditional finite differential method and finite element method. The loss function can be divided into several parts.

$$
L = L_F + L_I + L_B
\tag{2}
$$

where

$$
\begin{aligned}
L_F &= \frac{1}{n_F} \sum_{i=1}^{n_F} |\partial_t u_{net}(t_i^F, x_i^F) + L_x u_{net}(t_i^F, x_i^F) - f(t_i^F, x_i^F)|^2 \\
L_I &= \frac{1}{n_I} \sum_{j=1}^{n_I} |u_{net}(0, x_j^I) - g(x_j^I)|^2 \\
L_B &= \frac{1}{n_B} \sum_{k=1}^{n_B} |u_{net}(t_k^B, x_k^B) - h(t_k^B, x_k^B)
\end{aligned}
\tag{3}
$$

In PINN, $(t_i^F, x_i^F), x_j^I$ and $(t_k^B, x_k^B)$ are randomly sampled from a given distribution. Specially, $u(t, x)$ is derived from a back-propagation neural network by minimizing the proposed loss function 2

$$
\theta = \underset{\theta}{\arg\min} \, L
\tag{4}
$$

### 2.2 SELF-PACED LEARNING

Self-paced Learning(Kumar et al., 2010) is one of the significant branches of Automatic Curriculum Learning(Bengio et al., 2009), which devotes a readily computable example-wise the training loss as Difficulty Measurer, alleviating the lack of how to efficiently identify easy samples.

Let $D = \{\mathbf{x}_i, \mathbf{y}_i\}_{i=1}^N$ be the training dataset where $\mathbf{x}_i \in \mathcal{X}$ represents the $i^{th}$ input variable and $\mathbf{y}_i \in \mathcal{Y}$ represents the correspondingly $i^{th}$ output variable. Given the training data, the parameters $\theta$ of a special model are trained by minimizing the defined loss function $L$. Traditionally, the parameters are trained as follows:

$$
\theta = \underset{\theta}{\min}\{\sum_{i=1}^N L(\mathbf{x}_i, \mathbf{y}_i; \theta) + R(\theta)\}
\tag{5}
$$

where $R(\cdot)$ is a Regularizer. SPL introduces SPL-Regularizer to the traditional learning pattern, thus parameters are trained in a meaningful way.

$$
\theta = \underset{\theta}{\min}\{\sum_{i=1}^N v_i L(\mathbf{x}_i, \mathbf{y}_i; \theta) + g(v_i, \lambda)
\tag{6}
$$

The value of each $v_i$ is in the interval [0, 1], $\lambda$ is a barrier which determines the number of easy samples that are selected at each training epoch. There are numerous types of SP-Reqularizer and the corresponding close-formed solutions $v^*(L, \lambda)$ such as Hard-SP-Reqularizer, Linear-SP-Reqularizer, Logarithmic-SP-Reqularizer, Mixture-SP-Reqularizer, Mixture2-SP-Reqularizer, Logistic-SP-Reqularizer, Polynomial-SP-Reqularizer.Wang et al. (2022) We choose to use Hard-SP-Reqularizer to illustrate the efficiency of our method in section 4.

## 3 THE SPL-PINN FRAMEWORK

While PINN has shown its remarkably success in solving a variety of forward and inverse partial differential equations, it ignored the learning efficiency of the model itself, which resulting in a great demand for an effective method to solve this defect. Enormous research output was devoted to this topic, mainly motivated by the different weights of losses. Admittedly, the performance and the weights of losses are inextricably interconnected. However, we try to approach the problem from a different viewpoint. Strategies for the difficulty in converging are an important component of PINN, to satisfy the demand, we propose the SPL-PINN framework, which can effectively alleviate the inefficiency of convergence. Given the training data, instead of pull it into PINN in an meaningless way, we choose to fit the model from easy one to hard one, which is eminently suitable for human learning process.

### 3.1 SPL-PINN LOSS FUNCTION

In order to accelerate the learning process, we determinate the loss function with a SPL-Regularizer. Samples whether are easy or not are determined by a scalar $\lambda$. In order to concisely exploit the SPL-PINN structure, we choose to use the Hard-SP-Reqularizer and its corresponding $v^*$ to control the portion of easy samples that are plugged into the model.

$$
L_{original} = L_F + L_I + L_B
$$

$$
L = \frac{1}{N} \sum_{i=1}^{N} v_i L_{original}(\mathbf{z_i}; \theta) + g(v_i, \lambda)
$$

$$
g(v_i, \lambda) = -\lambda \sum_{i=1}^{N} v_i \tag{7}
$$

$$
v_i^*(l_i; \lambda) = \begin{cases} 1, & l_i < \lambda \\ 0, & otherwise \end{cases}
$$

The specific details are as follows:

- $v_i = 1$ means that the $i^{th}$ sample is identified as the *easy* one, which should be used to train the model.
- $v_i = 0$ means that the $i^{th}$ sample is identified as the *hard* one, which should not be used to train the model.
- $\lambda$ is a threshold, the larger $\lambda$ means more samples will be selected.

### 3.2 OVERVIEW OF SPL-PINN

Normally, the dataset is constructed by randomly sampled from a given distribution. Generally, researchers regard all of the samples have equal contribution to their models. Considering that our deep learning models are finally being used to make effective predictions that solve partial differential equations, especially those equations have physical information, it seams meaningless to some degree to just treat samples equally.

In consideration of a human learning habit, which usually starts from simple and general ones then gradually transmits to more complex and specialized ones. For instance, college students normally learn mathematical analysis and algebra at the first semester, later they are available to access real

and complex analysis, functional analysis. Given the hypothesis that different samples play different role in a model training process. Motivated by the habit of human learning behavior, we propose a novel SPL-PINN framework, which seamlessly integrates PINN and Self-paced Learning.

We consider the partial differential equation with both inital-value condition and boundary-value condition that proposed in section 2.1. For the $i^{th}$ sample, firstly we compute its loss function $l_i$. Secondly, identifying whether it is an easy sample by $v_i^*(l_i, \lambda)$. Subdataset is constructed by selceted samples. Thirdly, model parameters $\theta$ are renewed by minimizing the loss function $L$. Finally, after converging in this subdataset, update the threshold $\lambda$. The complete procedures are as follows:

---

**Algorithm 1** SPL-PINN

---

**Input:** Input dataset $\{(t_i^F, x_i^F)\}_{i=1}^{n_F}$, $\{x_j^I\}_{j=1}^{n_I}$ and $\{(t_k^B, x_k^B)\}_{k=1}^{n_B}$; pace parameter $\sigma$
**Output:** Model parameter $\theta$
 1: Initialize $\theta$, $\lambda$;
 2: **while** not converged **do**
 3:     **while** not converged **do**
 4:         Compute $v_i^*(l_i, \lambda)$
 5:         Update $\theta = \underset{\theta}{\arg\min} L$
 6:     **end while**
 7:     $\lambda \leftarrow \sigma\lambda$
 8: **end while**

---

## 4 EXPERIMENTS

To illustrate the efficiency of our method in the converging process, we apply SPL-PINN to a typical parabolic equation and Burgers equation. The dataset is constructed by randomly sampled from the uniform distribution.

### 4.1 HEAT EQUATION

The heat equation, also known as diffusion equation, is one of typical parabolic equations. It is a widely studied topic and foundational partial diffusion equation, which describes the evolution in time of the density $u$ of some quantity such as heat.

$$
\begin{aligned}
&\frac{\partial u}{\partial t} - \frac{\partial^2 u}{\partial x^2} = 0 \quad (x, t) \in (0, 1) \times (0, 1)\\
&u(x, 0) = \sin(\pi x) \quad x \in (0, 1)\\
&u(0, t) = u(1, t) = 0 \quad t \in (0, 1)
\end{aligned}
\tag{8}
$$

The problem stated above has its unique solution:

$$
u(x, t) = \sin(\pi x)e^{-\pi^2 t}
\tag{9}
$$

We choose the training dataset $\{(t_i^F, x_i^F)\}_{i=1}^{n_F}$, $\{x_j^I\}_{j=1}^{n_I}$ and $\{(t_k^B, x_k^B)\}_{k=1}^{n_B}$ which $n_F = n_I = n_B = 2000$. For a fixed iteration, the proformance of SPL-PINN is better than traditional PINN. For a given training epoch, we compute mean squared error of $u(x, t)$ in test dataset. Mean squared errors of $u(x, t)$ of given epochs are represented in Table 1. Moreover, the solution of 8 and the train losses of PINN and SPL-PINN can be found in 12. We use $\delta$ to measure the residual, $\delta = L_{\text{SPL-PINN}} - L_{\text{PINN}}$. Typically, $\delta$ is almost everywhere negative, which is a compelling illustration of the efficiency of our method.

### 4.2 BURGERS EQUATION

Burgers equation expresses the propagation and reflection of shock wave. Naturally, it plays a significant role in many applications such as gas dynamics, fluid mechanics, nonlinear acoustics and so on. The equations are as follows:

Table 1: Mean squared error of $u(x,t)$

| epochs | PINN | SPL-PINN |
|--------|--------|----------|
| 10 | 0.2016 | 0.1912 |
| 100 | 0.1322 | 0.1077 |
| 200 | 0.0421 | 0.0406 |
| 300 | 0.0388 | 0.0167 |

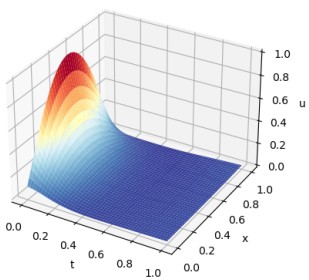

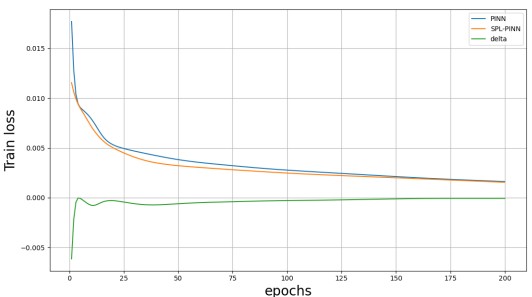

Figure 1: the solution $u(x,t)$ of heat eqution by SPL-PINN

Figure 2: train losses of PINN and SPL-PINN

$$\frac{\partial u}{\partial t} + u\frac{\partial u}{\partial x} - \omega\frac{\partial^2 u}{\partial x^2} = 0 \quad (x,t) \in (-1,1) \times (0,1)$$
$$u(0,x) = -\sin(\pi x) \quad x \in (-1,1)$$
$$u(t,-1) = u(t,1) = 0 \quad t \in (0,1) \tag{10}$$
$$\omega = \frac{0.01}{\pi}$$

We choose the training dataset $\{(t_i^F, x_i^F)\}_{i=1}^{n_F}, \{x_j^I\}_{j=1}^{n_I}$ and $\{(t_k^B, x_k^B)\}_{k=1}^{n_B}$ which $n_F = n_I = n_B = 2000$. Given the training epoch, the mean squared error is presented in Table 2 3 while the solution of 10 solving by SPL-PINN is presented in Figure 3. In addition, the solution of 10 and the train losses of PINN and SPL-PINN can be found in 34. Obviously, the great portion of $\delta$ is under 0.

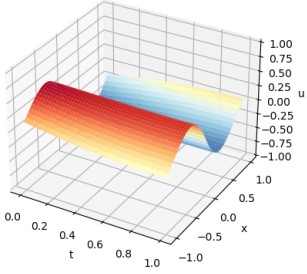

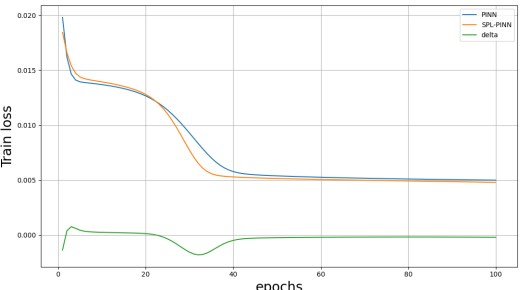

Figure 3: the solution $u(x,t)$ of Burders eqution by SPL-PINN

Figure 4: train losses of PINN and SPL-PINN

Table 2: Mean squared error of the initial condition for given epochs

| epochs | PINN(initial) | SPL-PINN(initial) |
|---|---|---|
| 10 | 0.5592 | 0.5499 |
| 100 | 0.2564 | 0.2464 |
| 200 | 0.2363 | 0.2519 |
| 250 | 0.2290 | 0.2245 |

Table 3: Mean squared error of the boundary conditions for given epochs

| epochs | PINN($x = -1$) | SPL-PINN($x = -1$) | PINN($x = 1$) | SPL-PINN($x = 1$) |
|---|---|---|---|---|
| 10 | 0.1451 | 0.1317 | 0.1344 | 0.1274 |
| 100 | 0.0528 | 0.0339 | 0.0525 | 0.0458 |
| 200 | 0.0331 | 0.0416 | 0.0397 | 0.0408 |
| 250 | 0.0388 | 0.0237 | 0.0416 | 0.0205 |

## 5 CONCLUSION

Recently, PINN has gained high attention. There are many researchers devote their efforts on embellishing it. Admittedly, The difficulty in converging is a huge concern. In an attempt to solve the inefficiency of convergence, we propose a noval SPL-PINN framewrok. It is a new training pattern, samples plugged into the model in a meaning way, from easy ones to hard ones. The performance in hear equation and Burgers equation illustrate the effectiveness of this new pattern.

## 6 FUTURE WORK

Normally, the loss function of PINN is combined by three parts: the PDE residual, the initial condition residual and the boundary condition residual. Considering the three parts maybe in different magnitude, is it fair to put the equal weights to them? In future, we plan to propose a framework to guarantee the fairness of the weights of different losses.

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
