# OpenReview forum: "Self-Paced Learning  Enhanced Physics-informed Neural Networks for Solving Partial Differential Equations"
_ICLR.cc/2023/Conference — Submitted to ICLR 2023_

### Official Review · Reviewer_WN4N · 2022-10-20

**Confidence:** 5
**Clarity, Quality, Novelty And Reproducibility:** The paper is not well written and has…
**Correctness:** 2
**Technical Novelty And Significance:** 1
**Empirical Novelty And Significance:** Not applicable
**Recommendation:** 1

**Strength And Weaknesses:**

Strengths
- None.

Weaknesses
- The paper is not well written.
- The proposed SPL loss is essentially using a point-wise weight coefficient. Exact the same idea has been proposed in https://arxiv.org/abs/2009.04544 and https://arxiv.org/abs/2001.04860
- No sufficient examples.


**Summary Of The Paper:**

The paper proposed to accelerate the training of PINNs by self-paced learning (SPL). A heat equation and Burgers’ equation are used to test the performance of the proposed method.

**Summary Of The Review:**

The method proposed in this method has already developed in other papers. The numerical results are not well written, and the method is not well investigated.

---

### Official Review · Reviewer_kVD8 · 2022-10-23

**Confidence:** 4
**Correctness:** 3
**Technical Novelty And Significance:** 2
**Empirical Novelty And Significance:** 2
**Recommendation:** 3

**Clarity, Quality, Novelty And Reproducibility:**

The idea is simple and easy to follow. There are too many typos in the writing, and the format is not ready for publish.

**Strength And Weaknesses:**

Strength:
1. The paper focuses on an emerging and important research area, the convergence of PINN.
2. The idea of introducing Self-paced Learning into PINN training is straightforward and interesting.


Weaknesses:
1. The format of the paper seems not ready for ICLR main conference.
2. The idea is too naive and the experimental part is too simple and lacks deep analyses.
3. There are too many typos.

**Summary Of The Paper:**

This paper introduces Curriculum Learning, specifically Self-paced Learning, into the physics-informed neural networks (PINNs) learning, so as to train the model from easy samples to hard ones to improve the convergence. The proposed method simple takes the example-wise training loss as Difficulty Measurer to determine which training sample should be used during the training. The effectiveness of the proposed method is verified in a typical parabolic equation and Burgers equation.

**Summary Of The Review:**

This paper seems not to be ready.

---

### Official Review · Reviewer_mRmU · 2022-10-25

**Confidence:** 4
**Correctness:** 3
**Technical Novelty And Significance:** 2
**Empirical Novelty And Significance:** 2
**Recommendation:** 3

**Clarity, Quality, Novelty And Reproducibility:**

The method and introduction part of the paper is well written. The experiment part is kind of rushed.

**Strength And Weaknesses:**

Weaknesses:
- The experiments are too toy to show the true efficacy of the proposed method. Also, the hyperparameters of the experiments are hidden, which makes the result uninterpretable.
- My main reservation is about the novelty. The proposed method does not contain significant new techniques. In my opinion, I would like to call it a trick instead of a scientific discovery.

**Summary Of The Paper:**

This manuscript proposes a method that can adaptively scale the loss based on the error of predicting the results of PDE. This manuscript experimented with the method on several classic PDEs, including the heat equation and burger's equation.

**Summary Of The Review:**

This submission requires significant work based on the evaluation above.

---

### Official Review · Reviewer_VxbH · 2022-10-25

**Confidence:** 4
**Correctness:** 1
**Technical Novelty And Significance:** 2
**Empirical Novelty And Significance:** 1
**Recommendation:** 1

**Clarity, Quality, Novelty And Reproducibility:**

# clarity

the paper is in a draft stage, the writing is not polished at all

# quality

it is a work in progress, the equations and variables are not well defined or explained

# novelty

using curriculum learning with PINNs seems to be a novel idea

# reproducibility

it is not reproducible, no code, and no details of experiments are given

**Strength And Weaknesses:**

# Strengths

- good results

#Weaknesses

- badly written
- limited experiments (only two toy problems)
- equations unclear and not well explained
- the "proposed loss function" (eq. 4) is completely trivial
- references are incomplete (no authors are given for the main reference)
- unclear what lambda is

**Summary Of The Paper:**

PINNs have difficulty with convergence. Data is usually randomly sampled according to the authors. This is why the authors propose to use a better sampling strategy.

**Summary Of The Review:**

The paper is an incomplete draft

---

### Decision · Program_Chairs · 2023-01-20

**Decision:**

Reject

**Justification For Why Not Higher Score:**

All the reviewers raised their concerns on the quality of the paper, on both technical innovation, experiments, and even paper writing. It looks like an incomplete draft and clearly not ready for ICLR. The authors did not provide their rebuttal.


**Justification For Why Not Lower Score:**

N/A

**Metareview: Summary, Strengths And Weaknesses:**

This manuscript proposes a method that can adaptively scale the loss based on the error of predicting the results of PDE. This manuscript experimented with the method on several classic PDEs, including the heat equation and burger's equation.

All the reviewers raised their concerns on the quality of the paper, on both technical innovation, experiments, and even paper writing. It looks like an incomplete draft and clearly not ready for ICLR. The authors did not provide their rebuttal.